# Spatial Path Planning for Robotic Milling of Automotive Casting Components Based on Optimal Machining Posture

**Hao Wu** [1,2], **Yudi Wang** [1,2], **Xiaoxu Wei** [1,2,*] and **Dahu Zhu** [1,2]

1 Hubei Key Laboratory of Advanced Technology for Automotive Components,
Wuhan University of Technology, Wuhan 430070, China; wuhao20228@163.com (H.W.);
wyudi2022@163.com (Y.W.); dhzhu@whut.edu.cn (D.Z.)
2 Hubei Collaborative Innovation Center for Automotive Components Technology,
Wuhan University of Technology, Wuhan 430070, China
* Correspondence: wxx2014@whut.edu.cn; Tel.: +86-27-87858200

**Abstract:** The robotic milling of automotive casting components can effectively reduce human participation in the production process and enhance production efficiency and quality, but the premise addresses the reasonable planning of machining paths. To address major challenges, this paper proposes a spatial path planning method for the robotic milling of casting flash and burrs on an automotive engine flywheel shell based on the optimal machining posture. Firstly, an improved stereolithography slicing algorithm in arbitrary tangent plane direction is put forward, which solves the problem that the existing stereolithography slicing algorithm cannot accurately extract the contour of complex components. Secondly, the contour path curve fitting of the slicing points of the flywheel shell is realized based on the B-spline curve. Next, a machining posture evaluation function is established based on the robot's stiffness performance, and the optimal machining posture is solved and verified with simulation according to the machining posture evaluation function and posture interpolation. Finally, the experiments indicate that the proposed method can significantly enhance the machining quality, with an average allowance height of 0.33 mm, and reduce the machining time to 9 min, compared with the conventional manual operation, both of which satisfy the machining requirements.

**Keywords:** robotic milling; path planning; automotive components; slicing algorithm; robot's stiffness performance; machining posture



## 1. Introduction

Robotic machining, as an advanced manufacturing technology [1] that conforms to the national situation [2,3], receives wide concerns in recent years due to its advantages in improving the level of automation, as well as manufacturing efficiency and quality. This type of automated technology has been extensively used in the fields of aerospace and rail transit but poses major challenges in the automotive industry. For example, removing flash and burrs on automotive casting components such as engine flywheel shells still depends on manual operation in automotive enterprises, exhibiting the typical problems of low efficiency, poor consistency, and high labor intensity. This situation, however, is being changed by the robotic machining mode by virtue of flexibility, reconfigurability, and cost-effectiveness. The premise for the robotic milling of such complex casting components lies in spatial path planning, including the machining path generation [4,5], the stereolithography (STL) slicing algorithm, and the optimal posture solution.

For the complex automotive component of flywheel shells, reasonably planning the spatial path is a primary task in robotic machining processes such as milling. Currently, many methods are put forward to generate machining paths based on complex surface features, for instance, the section plane method [6], the parameter method [7], and the equal residual height method [8]. Due to the complex structure of flywheel shells, machining

paths are distributed on multiple free-form surfaces. As the section plane method [9,10] intercepts the machining surface by setting a set of equidistant section planes, it is suitable for the task requirements of this paper.

The section plane method usually takes a triangular mesh model as the intercepting object. STL is considered one of the most widely used file formats for triangular mesh models. The STL slicing algorithm based on the section plane method [11] obtains the intersection point by intersecting the section plane and the triangular patch, which can ensure the accurate extraction of path fitting points [12,13]. However, most STL slicing algorithms perform hierarchical slicing in the height direction and can only obtain the section profile of a certain height, which is difficult for the generation of contour paths for complex components. Aiming at the requirement of the contour path extraction of complex components such as flywheel shells, this paper proposes an arbitrary tangent-plane-based STL slicing algorithm to obtain slicing points and generates the task path by fitting the slicing points with a B-spline curve. This method has strong versatility and can also extract the machining path for other complex components.

In terms of robotic machining posture optimization, the majority of the existing methods eliminate the redundant degrees of freedom in the robotic machining process by adding constraints or optimization indicators. Based on the dexterity index, Xiao et al. [14] and Zhu et al. [15] established a multi-objective optimization model for the elimination of redundant degrees of freedom of machining robots and integrated the joint avoidance limit index and the robot dexterity index to complete robot pose optimization. By virtue of the stiffness performance index, Xiong et al. [16] proposed a new frame invariant performance index; Chen et al. [17] used the inverse of the volume of compliance ellipsoids as the stiffness performance index; Sun et al. [18] proposed the measured stiffness performance evaluation metrics that explain the difference in stiffness in two opposite directions. In addition, the constraints that comprehensively consider multiple specific indicators have also been proposed. For example, Chen et al. [19] proposed a method to achieve posture optimization by controlling the functional redundancy of the robot, which comprehensively considers the deformation caused by the spindle weight and the deformation caused by the cutting force. Then, the cutting trajectory of the robot is optimized with the kinematic performance index as the optimization goal. For the robotic milling system in this paper, the robot should avoid singularity and joint overrun and ensure the maximum stiffness performance.

The studies above indicate that the existing STL slicing algorithms can only obtain the cross-sectional contour of a certain height, which cannot meet the extraction requirements for the complex and special-shaped flywheel shell. Therefore, this paper aims to develop an improved STL slicing algorithm for reasonable and effective robotic milling path planning. To achieve this objective, in Section 2, an improved STL slicing algorithm in an arbitrary direction is proposed to extract the slicing points of a flywheel shell, and the machining curve fitting is completed through a B-spline curve from the slicing points. Meanwhile, in Section 3, a weighted posture evaluation function is established to optimize the robotic posture by considering robot singularity, joint avoidance limit, and stiffness performance. Through the path point posture interpolation, the minimum value of the posture evaluation function of each path point is obtained to realize the robotic milling posture optimization. Finally, in Section 4, the spatial path planning of the robotic milling of the engine flywheel shell is completed, and the superiority of the proposed method is verified with experiments.

## 2. STL-Based Path Planning for Robotic Milling of Flywheel Shells

### 2.1. Contour Path Segment Based on Geometric Features of the Flywheel Shell

STL is a file format for describing three-dimensional graphics [20], which represents the three-dimensional graphics as several triangular patches. Before acquiring the STL file of the flywheel shell, it is necessary to determine the contour curve segment according to its features and task requirements. As shown in Figure 1, the flywheel shell's flash and burrs are mainly distributed on the upper edges, and the geometric features of the task path are

characterized by straight lines, curves, and circles. Accurately extracting the contour curve of the flywheel shell is critical to task path generation.

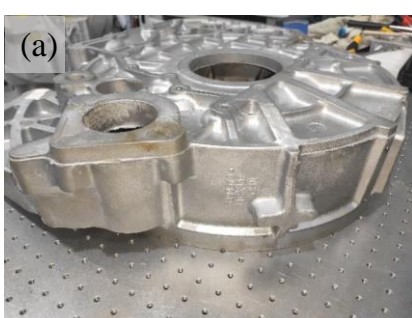
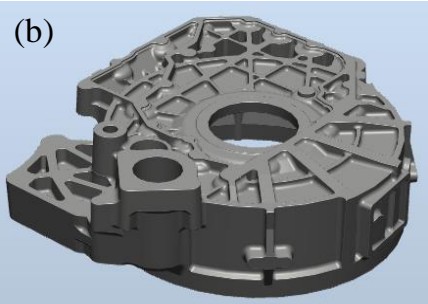

**Figure 1.** The automotive engine flywheel shell with flash/burrs (**a**) and the corresponding geometric model (**b**).

The contour curve is extracted based on the section plane method, and the task path is divided into horizontal plane and non-horizontal plane areas according to the geometric features of the contour. In Figure 2, both the green task paths and yellow circular task paths are classified into horizontal plane paths, while the red task paths are classified into non-horizontal plane paths with vertical planes and inclined planes. It can be seen that the red task paths divide the green ones into seven sections. A total of 10 horizontal plane paths exist with different heights, and their actual positions are shown in Figure 3.

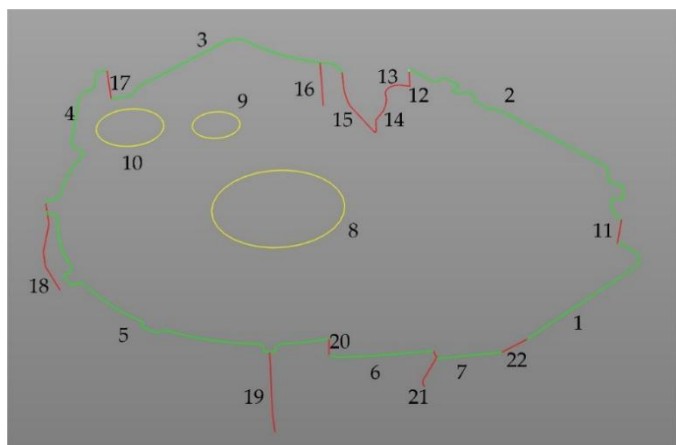

**Figure 2.** The task path distributions of the flywheel shell.

Table 1 further shows the plane heights of these 10 horizontal plane paths. According to the height values, the plane equation can be obtained, so the section plane of the task paths on the horizontal plane is given by $z = -h$.

**Table 1.** The path heights on the horizontal plane.

| Path Segment | Height $h$ (mm) | Path Segment | Height $h$ (mm) |
| --- | --- | --- | --- |
| 1 | 42.24 | 6 | 42.19 |
| 2 | 11.47 | 7 | 53.00 |
| 3 | 42.79 | 8 | 15.02 |
| 4 | 3.99 | 9 | 18.81 |
| 5 | 13.94 | 10 | 15.48 |

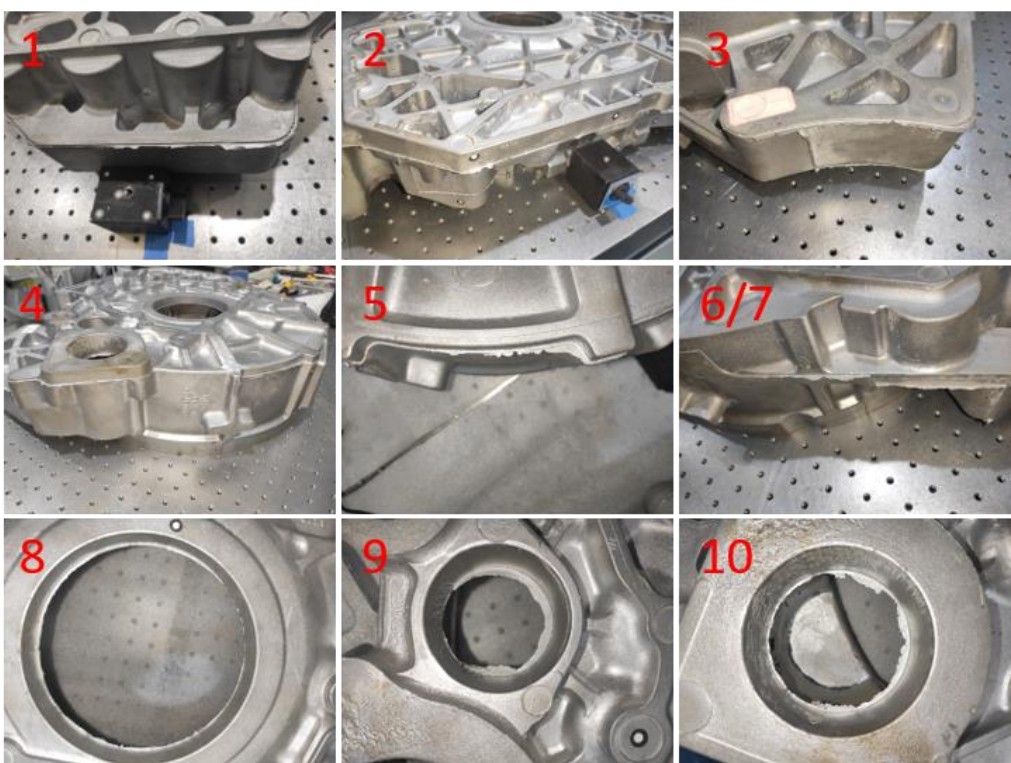

**Figure 3.** The actual positions of the task path segments on the horizontal plane.

By contrast, for the non-horizontal plane, the plane equation needs to be determined by three points that are not collinear. The coordinates of the three points that are not collinear can be taken from the geometric model and correspond to the actual positions. For a straight-line segment, the two ends of the line segment and the center of the circle are selected to construct the section plane, as shown in Figure 4. The three-point coordinates of each red task path segment are listed in Table 2.

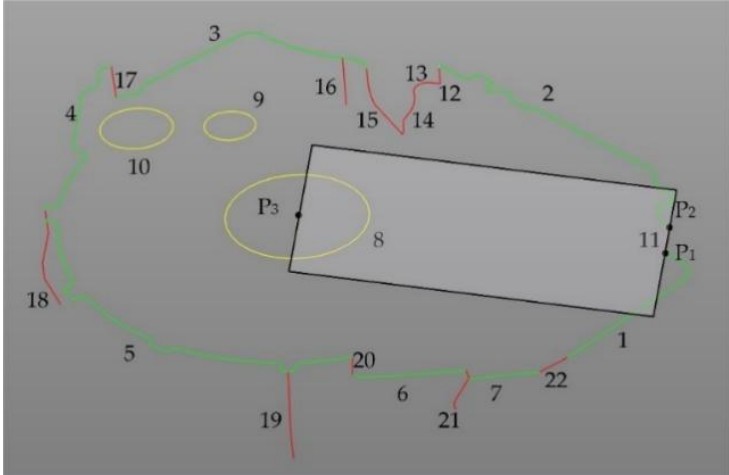

**Figure 4.** The section plane constructed by the three points that are not collinear.

**Table 2.** The three-point coordinates of non-horizontal path segments.

| Path Segment | $P_1$ | $P_2$ | $P_3$ |
|---|---|---|---|
| 11 | [285.43 232.34 −42.24] | [286.66 236.66 −11.47] | [0 0 0] |
| 12 | [−90.04 291.47 −11.47] | [−90.00 292.77 −36.23] | [0 0 0] |
| 13 | [−90.00 292.77 −36.23] | [−99.32 290.04 −37.86] | [−98.01 261.15 −38.10] |
| 14 | [−98.01 261.15 −38.10] | [−90.09 248.40 −49.44] | [−90.50 236.58 −56.69] |
| 15 | [−90.50 236.58 −56.69] | [−126.40 248.00 −76.57] | [−176.21 265.52 −42.79] |
| 16 | [−211.39 263.10 −42.79] | [−207.61 261.80 −101.79] | [0 0 0] |
| 17 | [−331.62 40.66 −42.79] | [−332.55 39.90 −3.99] | [0 0 0] |
| 18 | [−176.69 −176.69 −3.99] | [−183.14 −183.14 −68.25] | [−173.85 −173.84 −121.67] |
| 19 | [176.44 −176.44 −13.94] | [183.11 −183.11 −67.87] | [173.87 −173.63 −123.11] |
| 20 | [204.84 −120.39 −13.94] | [205.16 −121.93 −34.06] | [0 0 0] |
| 21 | [270.55 −36.74 −42.19] | [272.46 −34.26 −51.19] | [245.06 −24.94 −111.18] |
| 22 | [313.87 19.71 −52.98] | [327.15 43.81 −42.24] | [0 0 0] |

Given the coordinates of three points that are not collinear, $P_1(x_1, y_1, z_1)$, $P_2(x_2, y_2, z_2)$, $P_3(z_3, y_3, z_3)$, the plane equation passing through the three points is given by:

$$AX + BY + CZ + D = 0 \tag{1}$$

where the coefficients $A$, $B$, $C$, and $D$ of the plane equation are represented by the coordinates of the three points $P_1$, $P_2$, and $P_3$, respectively:

$$A = \begin{vmatrix} 1 & y_1 & z_1 \\ 1 & y_2 & z_2 \\ 1 & y_3 & z_3 \end{vmatrix}, B = \begin{vmatrix} x_1 & 1 & z_1 \\ x_2 & 1 & z_2 \\ x_3 & 1 & z_3 \end{vmatrix}, C = \begin{vmatrix} x_1 & y_1 & 1 \\ x_2 & y_2 & 1 \\ x_3 & y_3 & 1 \end{vmatrix}, D = \begin{vmatrix} x_1 & y_1 & z_1 \\ x_2 & y_2 & z_2 \\ x_3 & y_3 & z_3 \end{vmatrix}$$

In addition, due to the small number of triangular patches in the STL file exported by 3D modeling software such as SolidWorks (Solidworks2020, SolidWorks, Concord, MA, USA), the number of fitting points needs to be increased to ensure the accuracy of the path point fitting curve. Therefore, mesh subdivision is performed to ensure that the mesh density is in the same order of magnitude as the minimum distance of 1 mm from the path point in robot offline programming software such as RobotStudio. Finally, the STL file of the flywheel shell model is exported. The mesh subdivision is shown in Figure 5.

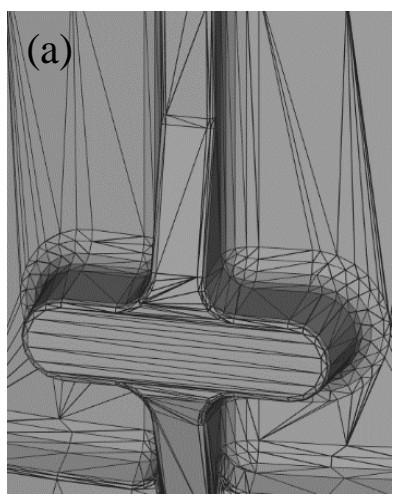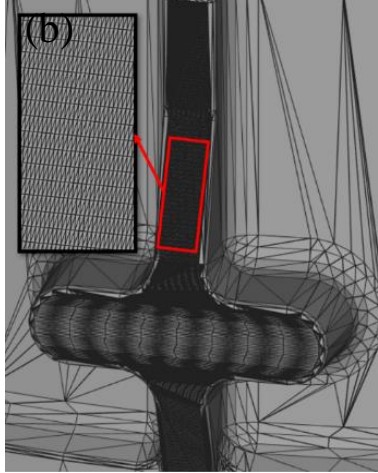

**Figure 5.** Comparison of before (**a**) and after (**b**) mesh subdivision of the flywheel shell STL file.

*2.2. STL Slicing Algorithm in an Arbitrary Direction Based on a Complex Parting Surface*

The STL slicing algorithm is a commonly used printing method in the field of 3D printing. However, most of the current STL slicing algorithms perform layered slicing in the height direction and can only obtain the cross-sectional contour of a certain height. Thus, it is difficult to generate the contour path of complex components such as flywheel shells. Therefore, the STL slicing algorithm needs improvement. In this paper, the method

of generating the tangent plane in an arbitrary direction is proposed accordingly, and the constraint of eliminating the redundant inner and outer contours is established. The detailed steps are as follows:

Step 1. Determine the tangent plane equation in an arbitrary direction.

The tangent plane of the task path of the flywheel shell has great randomness, and the tangent plane equation must be determined first. It is known that the three points that are not collinear can determine a plane, and the three points that are not collinear are arbitrarily selected in each path of the flywheel shell. For the straight-line segment path, the two endpoints of the path and the center of the flywheel housing circle are selected, and the section plane equation is expressed as

$$
\begin{cases}
Z = h & \text{horizontal plane} \\
AX + BY + CZ + D = 0 & \text{non-horizontal plane}
\end{cases}
\tag{2}
$$

Step 2. Determine whether the triangular patch intersects with the tangent plane.

In the tangent plane equation, the normal vector $N(A, B, C)$, any point on the tangent plane, and the three vertices of the triangular patch constitute three vectors. The inner product of the three vectors with the normal vector $N$ of the tangent plane can obtain three values. If a value exists that equals 0 among the three values, it means that the tangent plane passes through the vertices in the triangular patch, and no intersection occurs. If the three values are all positive or all negative, it means that the three vertices of the triangular patch are on the same side of the plane; that is, the triangular patch does not intersect with the tangent plane. Even if the three values are not all positive or all negative, it means that the three vertices of the triangular patch are located on both sides of the plane; that is, the triangular patch intersects with the tangent plane.

As shown in Figure 6, the three vertices of the triangular patch are defined as $P_4$ $(x_4, y_4, z_4)$, $P_5$ $(x_5, y_5, z_5)$, and $P_6$ $(x_6, y_6, z_6)$, and the three vertices constitute the vectors $P_1P_4$, $P_1P_5$, and $P_1P_6$ with the point $P_1$ on the tangent plane; thus, the judgment formula is written as

$$
\begin{aligned}
(\overrightarrow{P_1P_4} \cdot N) \wedge 0; \\
(\overrightarrow{P_1P_5} \cdot N) \wedge 0; \\
(\overrightarrow{P_1P_6} \cdot N) \wedge 0
\end{aligned}
\tag{3}
$$

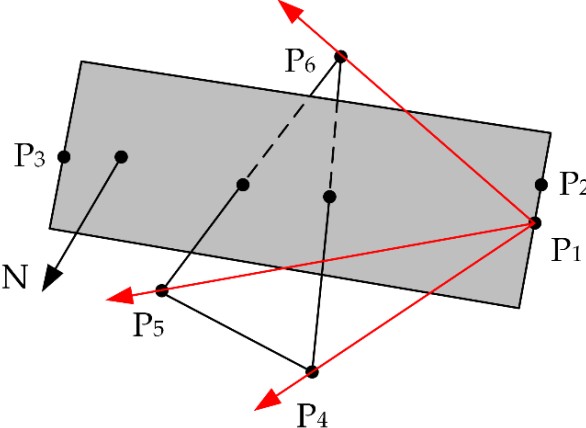

**Figure 6.** Intersection of the triangular patch with the section plane.

Step 3. Find the intersection point of the triangular patch with the tangent plane.

According to the distribution of the vertices of the triangular patch on the tangent plane in the previous step, the triangular patch that intersects with the tangent plane must have three vertices distributed on both sides of the tangent plane. This means that one vertex is located on one side of the tangent plane, and the other two vertices are located on the

other side of the tangent plane. By drawing a straight line from one vertex to the other two vertices, the straight-line equation is established according to the vertex coordinates and then is combined with the tangent plane equation to obtain the intersection coordinates.

Assuming that $P_1P_4$ and $P_1P_5$ are greater than zero, $P_1P_6$ is less than zero, and the intersections of the triangular patch with the section plane are $P_4P_6$, $P_5P_6$, then the straight-line equation is shown below:

$$\frac{X-X_4}{X_6-X_4} = \frac{Y-Y_4}{Y_6-Y_4} = \frac{Z-Z_4}{Z_6-Z_4}$$
$$\frac{X-X_5}{X_6-X_5} = \frac{Y-Y_5}{Y_6-Y_5} = \frac{Z-Z_5}{Z_6-Z_5} \quad (4)$$

Combined with the section plane Equation (2), the intersection points are acquired, respectively.

Step 4. Eliminate the inner contours.

By traversing all the triangular patches, the obtained intersections constitute the cross-sectional contour between the tangent plane and the flywheel shell, which includes the task path segments, as well as the possible useless inner and outer contours. Therefore, in order to eliminate the inner contours, two measures are taken. Firstly, the flywheel shell model is filled as a solid body to eliminate the inner contour, and secondly, for the contours outside the scope of the task path segment, the desired slicing points are filtered out according to the coordinates of the starting and ending points of the task segment, as well as the given definition domain of X and Y. In this way, the cross-sectional contour of the specified area model and the tangent plane can be solved.

### 2.3. Path Fitting Based on the B-Spline Curve Slicing Point

By using the STL slicing algorithm, the slicing points of the flywheel shell contour can be acquired, and the path curve is fitted accordingly. As the fitting curve needs to pass through all the slicing points, it is suitable for the spline curve in the field of computational geometry. The common splines include Bezier spline, B spline, and NURBS spline. Compared with the simple Bezier spline and complicated NURBS spline, the B spline can meet the fitting requirements and is, therefore, selected to fit the task path curve in this paper.

Before the curve fitting, the slicing points need to be sorted. Since the set of path points of each segment is ordered, it is only necessary to consider the connection order between segments. The path sequence is finally determined as 1→11→2→12→13→14→15→3→17→4 →5→20→6→7→22. The rest of the paths have no order requirement and can be fitted individually. In addition, denser slicing points are obtained through the above mesh subdivision. As shown in Figure 7, the slicing point spacing is significantly reduced by increasing the initial 1400 slicing points to 12,000.

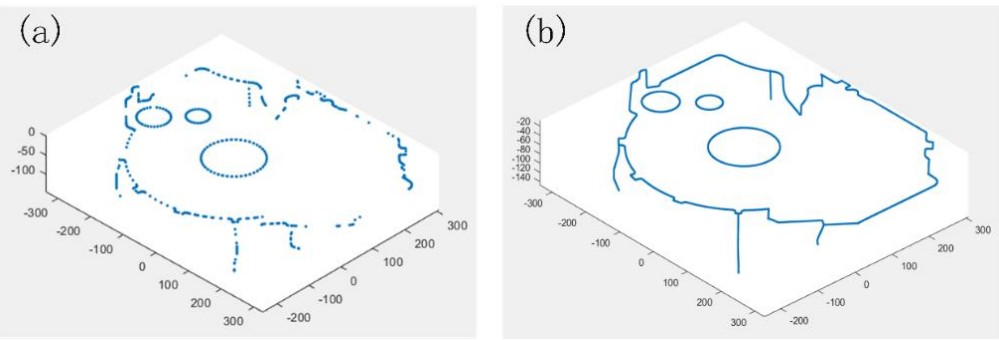

**Figure 7.** Comparison of the slicing point sets before (**a**) and after (**b**) preprocessing of the flywheel shell STL file.

After completing the above operation, the 15 segment paths are fitted in sequence. The equation of the B-spline curve is written as follows [21]:

$$p(u) = \sum_{i=0}^{n} d_i N_{i,k}(u) \tag{5}$$

Note that the B-spline curve is composed of the cumulative sum of multiple B-spline basis functions $N_{i,k}(u)$, and the shape of the curve is controlled by the control point $d_i$. The second subscript $k$ of the B-spline basis function $N_{i,k}(u)$ represents the degree of the curve, and $i$ is the serial number from 0 to $n$. The argument $u$ of the basis function is the parameter in the node vector $U$, and the parameters in the node vector $U$ are arranged in non-decreasing order, from 0 to $n + k + 1$. The fitting principle is as follows:

Step 1. Determine the node vector.

Assuming that $m + 1$ data points $q_i$ ($I = 0, 1, \ldots, m$) are obtained, a cubic general non-uniform B-spline curve should be constructed. First, it should be determined whether the curve to be constructed is an open curve or a closed curve; that is, whether the data points at the beginning and end are the same. If they are, it is a closed curve; otherwise, it is an open curve. For a closed curve, if $C^2$ is not required to be continuous at the overlap, it is equivalent to an open curve. The data point at the closure is used as the beginning and ending point of the curve, and the other data points are used as intermediate segment points so that the curve has a total of $m$ segments. The number of spline control points is equal to the number of data points plus the degree of the curve minus one. Assuming that there are $n + 1$ control points $d_j$ ($j = 0, 1, \ldots, n$), it has $n = m + 2$. Since the degree of the curve is 3 times, and the repeatability of the domain endpoint is 4, the standard domain $u \in [u_3, u_{n+1}] = [0,1]$. Thus, the endpoints with a repeatability of 4 at the beginning and ending are given by

$$u_0 = u_1 = u_2 = u_3 = 0$$
$$u_{n+1} = u_{n+2} = u_{n+3} = u_{n+4} = 1$$

For the standard definition domain of $m + 1$ data point $q_i$, the normative accumulation chord length method is used to calculate the chord length of $m + 1$ data points connected in turn. Assuming that the $m$-th chord length is $l_m$, the total chord length $L$ is expressed as

$$L = \sum_{1}^{m} l_m \tag{6}$$

According to the parameter sequence $u_i$ ($i = 0, 1, \ldots, m$) of the node vector, it has

$$\begin{cases} \_u_i = \dfrac{\sum_{1}^{i} l_i}{L}, & i = 1, 2, \ldots, m \\ \_u_0 = 0 \end{cases} \tag{7}$$

The correspondence between the standard domain and the node vector is $u_{3+i} = u_i$ ($i = 0, 1, \ldots, m$). For the $C^2$ continuous, closed curve in this paper, the standard definition domain is the same as above. The endpoints at both ends of the domain are set to

$$u_0 = u_{n-2} - 1, u_1 = u_{n-1} - 1, u_2 = u_n - 1, u_{n+2} = 1 + u_4, u_{n+3} = 1 + u_5, u_{n+4} = 1 + u_6$$

Step 2. Inversely calculate the control vertex.

The fitted curve equation can be written as

$$p(u) = \sum_{j=0}^{n} d_j N_{j,3}(u) = \sum_{j=i-3}^{i} d_j N_{j,3}(u), \quad u \in [u_i, u_{i+1}] \subset [u_3, u_{n+1}] \tag{8}$$

By substituting the node vector parameter $u \in [u_i, u_{i+1}] \subset [u_3, u_{n+1}]$ of the standard definition domain into the node vector determined by Step 1, the following interpolation conditions need to be satisfied:

$$
\begin{cases}
p(u_i) = \sum\limits_{j=i-3}^{i} d_j N_{j,3}(u_i) = q_{i-3}, & i = 3, 4, \ldots, n \\
p(u_{n+1}) = \sum\limits_{j=n-3}^{n} d_j N_{j,3}(u_{n+3}) = q_m
\end{cases}
\tag{9}
$$

Note that Equation (9) contains $n-1$ equations in total. Since $q_0 = q_m$, one equation is repeated; thus, there are $n-2$ after subtraction. Based on the definition of a closed curve, the three control points at the beginning and end should be the same, that is,

$$d_{n-2} = d_0, d_{n-1} = d_1, d_n = d_2$$

The required $n + 1$ control points are reduced to $n-2$, and the number of equations is equal to the number of unknowns, which is solved and represented by a matrix as follows:

$$
\begin{bmatrix}
N_{1,3}(u_3) & N_{2,3}(u_3) & & N_{0,3}(u_3) \\
N_{1,3}(u_4) & N_{2,3}(u_4) & N_{3,3}(u_4) & \\
\ddots & \ddots & \ddots & \\
& N_{n-4,3}(u_n) & N_{n-3,3}(u_n) & N_{n-2,3}(u_n) \\
N_{n-1,3}(u_{n+1}) & & N_{n-3,3}(u_{n+1}) & N_{n-2,3}(u_{n+1})
\end{bmatrix}
\begin{bmatrix}
d_1 \\ d_2 \\ \vdots \\ d_{n-3} \\ d_{n-2}
\end{bmatrix}
=
\begin{bmatrix}
q_0 \\ q_1 \\ \vdots \\ q_{n-4} \\ q_{n-3}
\end{bmatrix}
\tag{10}
$$

The matrix equation is further rewritten as

$$
\begin{bmatrix}
b_1 & c_1 & & a_1 \\
a_2 & b_2 & c_2 & \\
\ddots & \ddots & \ddots & \\
& a_{n-3} & b_{n-3} & c_{n-3} \\
c_{n-2} & & a_{n-2} & b_{n-2}
\end{bmatrix}
\begin{bmatrix}
d_1 \\ d_2 \\ \vdots \\ d_{n-3} \\ d_{n-2}
\end{bmatrix}
=
\begin{bmatrix}
e_1 \\ e_2 \\ \vdots \\ e_{n-3} \\ e_{n-2}
\end{bmatrix}
\tag{11}
$$

where

$$
\begin{aligned}
a_i &= \frac{(\Delta_i + 2)^2}{\Delta_i + \Delta_{i+1} + \Delta_{i+2}} \\
b_i &= \frac{\Delta_{i+2}(\Delta_i + \Delta_{i+1})}{\Delta_i + \Delta_{i+1} + \Delta_{i+2}} + \frac{\Delta_{i+1}(\Delta_{i+2} + \Delta_{i+3})}{\Delta_{i+1} + \Delta_{i+2} + \Delta_{i+3}} \\
c_i &= \frac{(\Delta_{i+1})^2}{\Delta_{i+1} + \Delta_{i+2} + \Delta_{i+3}} \\
e_i &= (\Delta_{i+1} + \Delta_{i+2})q_{i-1} \\
\Delta_i &= u_{i+1} - u_i \\
i &= 1, 2, \ldots, n - 2
\end{aligned}
$$

The control points can be acquired by solving the linear equations. By substituting the control points of the task path into Equation (5), the fitting curve is obtained as shown in Figure 8.

Based on these steps, the independent path segments are drawn in turn to obtain the complete task path. Finally, the complete task path is imported into the robot offline programming software (ABB RobotStudio) and matched with the flywheel shell model. As shown in Figure 9, the two fit well. Specifically, the distance values between the centers of the three circles on the trajectory are measured as 98.61 mm, 238.33 mm, and 185.75 mm, and then compared with the model, indicating the accuracy of the task path.

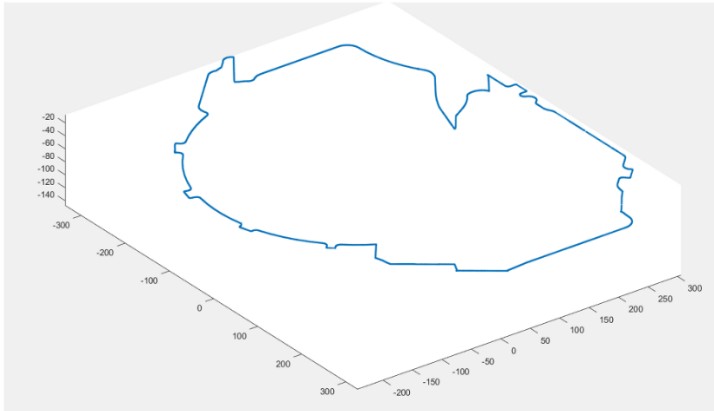

**Figure 8.** The fitting curve of the flywheel shell task path.

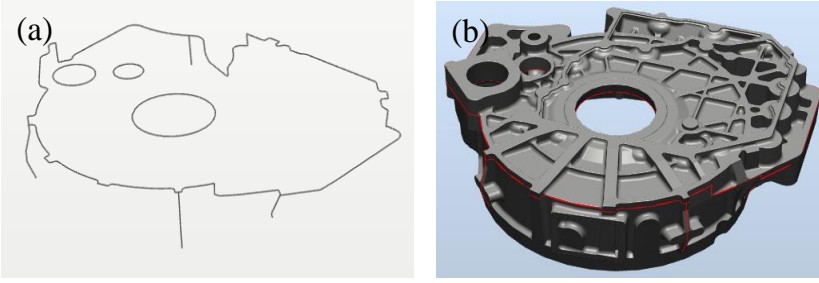

**Figure 9.** Comparison of the task path (**a**) with the standard model (**b**).

## 3. Optimal Machining Posture Solution Based on the Robot's Stiffness Performance

### 3.1. Machining Posture Evaluation Function Based on the Robot's Stiffness Performance

The robotic machining path includes the position and orientation (pose) information of the path point, in which the pose information is the frame at the path point. In robotic milling, the tool is vertically downward by default, and the frame can only be rotated around the tool axis to adjust the posture. Therefore, aiming at the redundancy of the robot's degrees of freedom, a path posture optimization function is proposed to solve the optimal posture of the endpoints of the path segment. Then, the intermediate points of the path are interpolated to complete the path planning.

Due to the large tangential and normal forces in the robotic milling process, better stiffness performance is required in these two directions. Therefore, the directional stiffness index $K_{ef}$ is used as follows [22]:

$$K_{ef} = \frac{1}{e_f^T C_{fd} e_f} \tag{12}$$

where $f$ is the magnitude of the force vector at the end, and $e_f$ is the direction of the force vector.

Set the directional stiffness index $K_E$ as

$$K_E = \frac{1}{\sqrt{K_X^2 + K_Y^2}} \tag{13}$$

where $K_X$ is the normal vector and $K_Y$ is the tangential vector, and the directional stiffness index is obtained from the compliance matrix.

$$\begin{bmatrix} d \\ \delta \end{bmatrix} = \begin{bmatrix} C_{fd} & C_{md} \\ C_{f\delta} & C_{m\delta} \end{bmatrix} \begin{bmatrix} f \\ m \end{bmatrix} \tag{14}$$

$$C_{fd} = \begin{bmatrix} c_{11} & c_{12} & c_{13} \\ c_{21} & c_{22} & c_{23} \\ c_{31} & c_{32} & c_{33} \end{bmatrix} \tag{15}$$

The directions of each coordinate axis at the end of the robot are [1,0,0], [0,1,0], and [0,0,1], respectively, and the stiffness in the direction of the end coordinate axis is obtained.

$$K_X = \frac{1}{c_{11}}; K_Y = \frac{1}{c_{22}}; K_Z = \frac{1}{c_{33}} \tag{16}$$

The directional stiffness index $K_E$ is normalized, and the stiffness posture evaluation function $E_{rigidity}$ is established, as shown in Equation (17).

$$E_{rigidity}(\theta) = \frac{K_E - K_{E\min}}{K_{E\max} - K_{E\min}} \tag{17}$$

An accessibility index is proposed for judging which axis configuration is the closest to the mechanical origin of the robot when multiple solutions and multiple axis configurations exist for a certain position. Therefore, the accessibility posture evaluation function $E_{reach}$ is defined as shown in Equation (18).

$$E_{reach}(\theta) = \begin{cases} \sum\limits_{i=1}^{6} \frac{|\theta_i - \theta_{0i}|}{(|\theta_i|_{\max} - \theta_{0i})}; & 0 < \frac{\theta_i - \theta_{i\min}}{\theta_{i\max} - \theta_{i\min}} < 0.9 \\ \infty; & others \end{cases} \tag{18}$$

where $\theta_{0i}$ is the $i$-th joint angle when the robot is at zero position, $\theta_{i\max}$ is the maximum value among all solutions of the $i$-th joint, and $\theta_{i\min}$ is the minimum value among all solutions of the $i$-th joint. This formula is finally normalized.

For the singularity index, there are three common singularities in a six-axis serial robot, namely the wrist joint singularity, the shoulder joint singularity, and the elbow joint singularity. Since the overall posture of the robot in the robotic milling experiment of the flywheel shell is downward, both the shoulder joint and elbow joint singularities are not considered. For the wrist joint singularity, changing the second and third axes can make the robot reach most of the positions. Note that the joint angle 5 may be zero during milling operation, which needs to be avoided.

The range of robot joint 5 is $-135°\sim135°$, and 90% of the range is taken as effective. The absolute value of the joint 5 should be better than $120°$. The singularity posture evaluation function $E_{singular}$ is shown in Equation (19).

$$E_{singular}(\theta) = \frac{||\theta_5| - 120|}{120} \tag{19}$$

The above posture evaluation functions affect the robotic milling performance, among which the robot's stiffness has the greatest impact on the stability of the milling process. According to the investigation by Huo and Baron [23], the weight of singularity is greater than that of accessibility. After several attempts in this study, the rigidity weight is set to 0.5, followed by singularity to 0.3 and accessibility to 0.2. The overall posture evaluation function is finally rewritten as follows:

$$E = min\left\{ 0.5\Delta E_{rigidity} + 0.3\Delta E_{singular} + 0.2\Delta E_{reach} \right\} \tag{20}$$

Note that the smaller $E$ value indicates a better machining posture.

### 3.2. Optimal Machining Posture Solution and Simulation Verification

In the process of robotic milling of the flywheel shell, the direction of the milling cutter is always vertically downward, and its tool coordinate system can be arbitrarily rotated

around the Z axis. Therefore, from the coordinates of the endpoints of the path segment, an infinite number of joint angle combinations can be obtained.

When the posture information of the two endpoints of a curve is known, the posture of the intermediate point can be obtained by interpolating the posture of the two endpoints. For the two endpoints *M* and *N* of any curve, the initial frame of the two points is firstly set, and the initial posture is taken into the posture evaluation function. Then, the initial frame is rotated around the Z axis. In this way, the evaluation value of all the frames around the Z axis is calculated. Finally, the frame with the smallest evaluation value is selected as the optimal posture, and the rotation angles $Ang_M$ and $Ang_N$ are recorded.

To interpolate the posture of the intermediate path points, the interpolation point angle is determined by the distance between the path points. The total length of the path segment *MN* is set as the sum of the path segments connected by the intermediate path points and is denoted as *Length*. The distance corresponding to each intermediate point is the sum of all previous line segments and is denoted as $S_i$. The *i*-th intermediate point angle $Ang_i$ between the path segment *MN* is given by

$$Ang_i = \frac{S_i}{Length} \times (Ang_N - Ang_M) + Ang_M \tag{21}$$

For the known coordinate frames of all path points, only the endpoints are verified. In order to avoid the problem with intermediate points, the posture of all intermediate points is inversely solved to check whether they are reachable, whether collisions occur, and whether they are singular. If the above problems exist, the point is set as the endpoint, the curve is divided into two sections, and the above steps are repeated. Based on the optimal posture solution, the feasibility of the algorithm is verified in the simulation software. The simulation results in Figure 10 show that the robot can run the path segments without collision according to the obtained optimal posture paths.

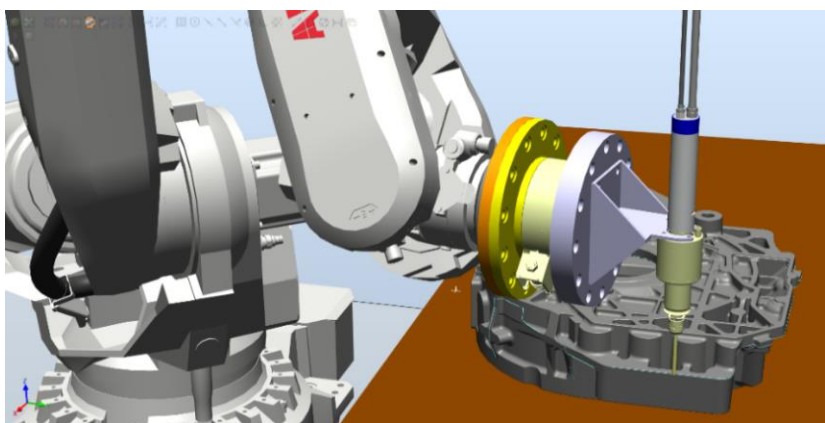

**Figure 10.** Simulation of the flywheel shell path segments.

## 4. Experimental Verification

### 4.1. Experimental Setup

Figure 11 shows the experimental platform for the robotic milling of the automotive engine flywheel shell flash and burrs. The experimental equipment mainly included an industrial robot (ABB IRB6700-200/2.60) and an automatic tool-changing electric spindle (NR4040-AQC). The workpiece to be machined was a flywheel shell with dimensions of 600 mm × 560 mm × 156 mm in length, width, and height, respectively. It was made of aluminum alloy 50021, and the corresponding chemical composition is listed in Table 3. The flywheel shell flash/burr to be removed was 0~5 mm in width and 1~2 mm in thickness. A four-edge tungsten steel milling cutter with a diameter of 10 mm, a length of 75 mm, a helix angle of 35°, and a maximum cutting hardness of HRC48 was used. The spindle

speed and the robot's feed speed were set as 8000 rpm and 5 mm/s, respectively, and the transition speed was set as 50 mm/s for efficiency.

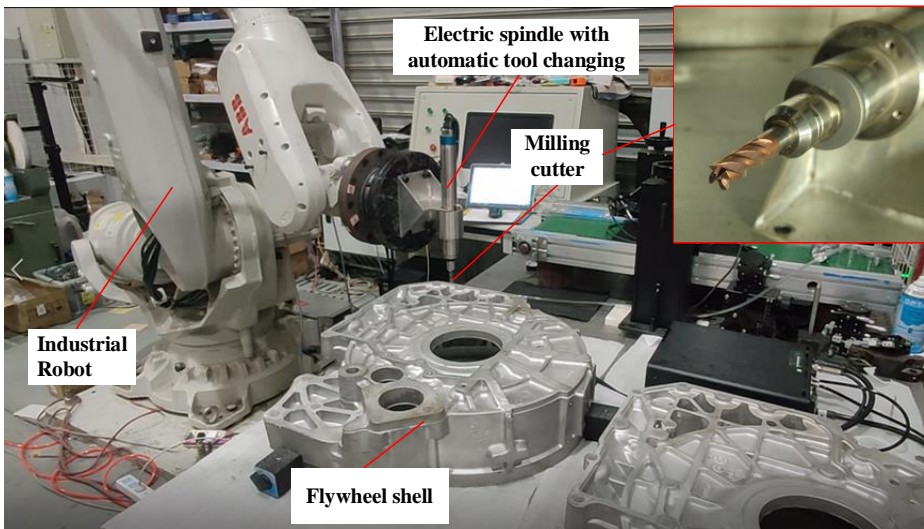

**Figure 11.** Experimental platform for robotic milling of the automotive engine flywheel shell flash and burrs.

**Table 3.** Chemical composition of the aluminum alloy 50021.

| Aluminum Alloy | Ingredient Content (%) | | | | | | | | |
|---|---|---|---|---|---|---|---|---|---|
| | Si | Fe | Cu | Zn | Mg | Mn | Pb | Sn | Ti + Zr |
| 50021 | 4.0–6.0 | ≤1 | 2.0–40 | ≤1 | ≤0.5 | ≤0.5 | ≤1 | ≤1 | ≤1 |

As shown in Figure 12, the path segments for planning the optimal machining posture are the plane segment paths in red, which accounts for 90% of the overall machining path. Thus, the red segment paths were used for experimental verification. Before performing the robotic milling operation, tool calibration and workpiece calibration had to be completed. The tool coordinate system was calibrated with the four-point method, while the workpiece coordinate system was calibrated with the three-point method [24]. Additionally, in order to ensure machining safety, it is necessary to verify the robotic milling path of the flywheel shell. The entire machining path height was enhanced by 3 mm with a slow running speed to observe the collision between the milling cutter and the flywheel shell.

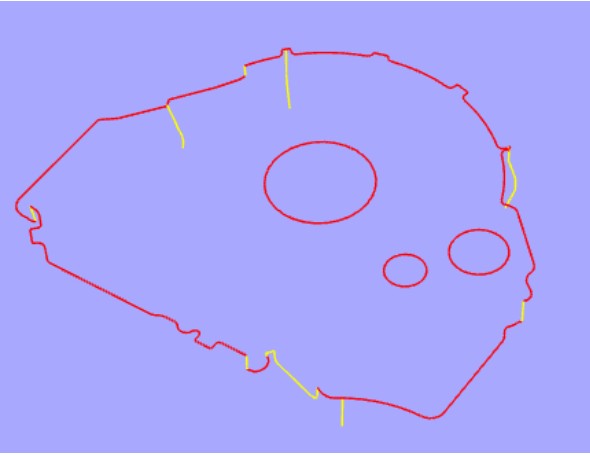

**Figure 12.** Robotic milling path of the flywheel shell in experiment.

### 4.2. Experimental Results and Analysis

According to the above experimental steps, the first straight-line segment, the third curved segment, and the tenth circular segment were selected as typical path segments for analysis. It can be seen in Figure 13 that the flywheel shell flash and burrs were greatly removed by the robotic milling in comparison with the manual operation.

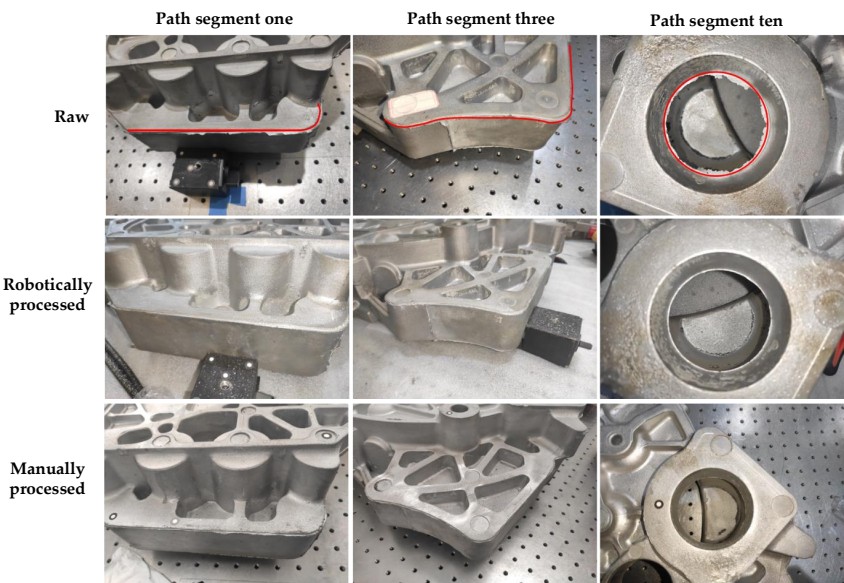

**Figure 13.** Comparison of the machining effects of flywheel shell flash/burrs by robotic milling and manual operation.

The machining effects were further quantified by measuring the allowance height of the flash/burrs on the straight-line (Str.), curved (Cur.), and circular (Cir.) segments with a dial indicator. Figure 14 shows the measurement process as follows: (1) The dial indicator was horizontally fixed on the lifting platform, and the needle vertically touched the machined area of the flywheel shell; (2) the height of the lifting platform was adjusted to measure the allowance height; (3) a total of 10 points were selected in each path for measurement, and the average value was taken to characterize the machining quality of each path. As listed in Table 4, the average allowance height after robotic milling was 0.33 mm, with high consistency, which was about half of the value of 0.65 mm by the conventional manual operation. Additionally, in terms of the machining efficiency, it took about 9 min for the robot to process 90% of the flywheel shell flash/burrs at one time, while the time is up to 15 min using manual operation. Thus, robotic milling enhanced the machining efficiency by about 40%.

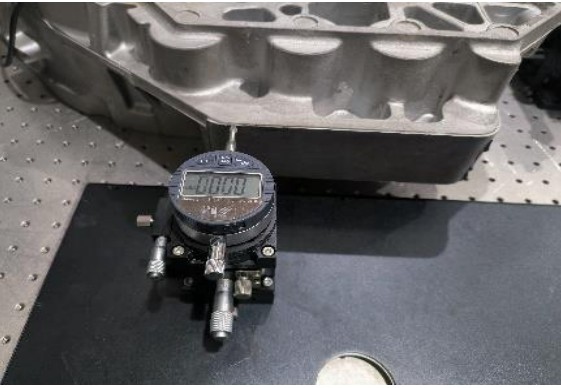

**Figure 14.** Allowance height measurement after machining.

**Table 4.** Comparison of the allowance height and time with two machining methods.

| Path segment | Robotic Milling | | | | Manual Operation | | | |
| | Allowance Height (mm) | | | Time (s) | Allowance Height (mm) | | | Time (s) |
| | Str. | Cur. | Cir. | | Str. | Cur. | Cir. | |
|---|---|---|---|---|---|---|---|---|
| 1 | 0.155 | 0.314 | 0.283 | 552 | 0.486 | 0.535 | 0.535 | 921 |
| 2 | 0.271 | 0.409 | 0.146 | 531 | 0.511 | 0.931 | 0.486 | 886 |
| 3 | 0.197 | 0.771 | 0.364 | 526 | 0.624 | 0.846 | 0.399 | 883 |
| 4 | 0.276 | 0.923 | 0.067 | 558 | 1.044 | 0.797 | 0.588 | 905 |
| 5 | 0.296 | 0.629 | 0.223 | 536 | 0.852 | 0.579 | 0.523 | 913 |
| 6 | 0.549 | 0.092 | 0.175 | 545 | 0.635 | 0.758 | 0.557 | 892 |
| 7 | 0.349 | 0.258 | 0.256 | 544 | 0.858 | 0.732 | 0.597 | 897 |
| 8 | 0.335 | 0.062 | 0.237 | 534 | 0.763 | 0.643 | 0.376 | 889 |
| 9 | 0.264 | 0.725 | 0.183 | 554 | 0.644 | 0.832 | 0.478 | 903 |
| 10 | 0.504 | 0.486 | 0.235 | 543 | 0.673 | 0.774 | 0.527 | 918 |

## 5. Conclusions

In this paper, a spatial path planning method for the robotic milling of an automotive engine flywheel shell's flash and burrs was proposed based on the optimal machining posture and then verified with experiments. The following conclusions were achieved:

(1) The improved stereolithography slicing algorithm could intercept the complex component contour in an arbitrary direction and eliminate the redundant points by separating the inner and outer contours under certain constraints. The generated task path accurately could match the model of the flywheel shell, showing excellent adaptability.

(2) The robotic posture evaluation function was established based on the stiffness performance, supplemented by the robot's collision, accessibility, and singularity indicators. The optimal machining posture of the path point was determined according to the frame rotation angle when the posture evaluation function value was the smallest.

(3) Both the simulation and experiment verified the feasibility of the proposed spatial path planning method. Compared with manual operation, the robotic milling of the flywheel shell's flash and burrs exhibited better machining quality and consistency. In particular, the machining time was reduced by 40%, compared with 15 min per piece by hand.

**Author Contributions:** Conceptualization, H.W. and Y.W.; methodology, H.W.; software, H.W.; investigation, H.W.; data curation, Y.W.; writing—original draft preparation, H.W.; writing—review and editing, X.W.; supervision, D.Z.; funding acquisition, D.Z. All authors have read and agreed to the published version of the manuscript.

**Funding:** This research was funded by the National Nature Science Foundation of China (No.51975443), and Hubei Province Key R&D Program (No. 2020BAA025).

**Institutional Review Board Statement:** Not applicable.

**Informed Consent Statement:** Not applicable.

**Data Availability Statement:** The raw/processed data required to reproduce these findings cannot be shared at this time, as the data also form part of an ongoing study.

**Conflicts of Interest:** The authors declare no conflict of interest.

## Nomenclature

| | |
|---|---|
| $Ang_i$ | Rotation angle of the *i*-th intermediate point |
| $C_{fd}$ | 3 × 3 order translation compliance matrix |
| $C_{f\delta}$ | 3 × 3 order coupling compliance matrix |

| | |
|---|---|
| $C_{md}$ | $3 \times 3$ order coupling compliance matrix |
| $C_{m\delta}$ | $3 \times 3$ order rotational compliance matrix |
| $d$ | Robot end coordinate system translation |
| $E$ | Overall posture evaluation function |
| $E_{reach}$ | Reachability posture evaluation function |
| $E_{rigidity}$ | Stiffness posture evaluation function |
| $E_{singular}$ | Posture evaluation function based on singularity point |
| $f$ | Force vector on the end of the robot |
| $K_E$ | Combined longitudinal and tangential directional stiffness |
| $K_{ef}$ | Directional stiffness index |
| $K_x$ | Normal stiffness |
| $K_y$ | Tangential stiffness |
| $m$ | Torque at the end of the robot |
| $STL$ | Stereolithography |
| $\delta$ | Robot end coordinate system rotation amount |

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
