# Peer review of "Spatial Path Planning for Robotic Milling of Automotive Casting Components Based on Optimal Machining Posture"

_metals, doi:10.3390/met12081271_

Round 1

Reviewer 2 Report

The reviewer comments of the paper

«Spatial Path Planning for Robotic Milling of Automotive Casting Components Based on Optimal Machining Posture»

- Reviewer

The authors presented an article «Spatial Path Planning for Robotic Milling of Automotive Casting Components Based on Optimal Machining Posture». However, there are several points in the article that require further explanation.

Comment 1:

The abstract needs to be improved.

Demonstrate in the abstract novelty, practical significance. Add quantitative and qualitative work results to the abstract.

Comment 2:

The introduction needs to be improved.

Firstly, group quotation is unacceptable in one phrase, for example [1-3], [11-13]. Break this sentence into parts or individual sentences. For example, ... [...], ... [...], etc. Or one reference - one sentence.

After analyzing the literature, show before formulating the goal of the "blank" spots. Which has not been previously done by other researchers. You must show the importance of the research being undertaken. Show what will be the new research approach in this article. You need to show a hypothesis.

Add scientific novelty and practical relevance.

Add a clear purpose to the article.

Briefly describe what is done in each section.

Comment 3:

2. STL based path planning for robotic milling of flywheel shell

Are all figures original? If not needed appropriate citations and permissions. Refine this for figures throughout the article.

Are all formulas original? If not needed appropriate citations.

Comment 4:

4. Experimental verification

Add the material chemistry of the stock in a separate table. What is the hardness of the workpiece and how was it measured?

Describe in table the geometry of the cutter used in the research (diameter, number of teeth, rake and clearance angle, etc.). Show these dimensions in the photo.

Show the direction of the machine axes. How does this compare to measured cutting forces? What kind of milling scheme is used? Describe in the text.

Describe the measurement procedure in more detail. At what point in time? How is the measuring setup set up? How many repetitions of measurements? What statistical methods are used to process experimental results? Describe the experimental stand in more detail. What method of experiment planning is used and why?

Comment 5:

It will be useful to add a section of Nomenclature in which to sign all the physical quantities and abbreviations encountered in the article. There are many physical quantities in the text and such a section will help to find the description of the necessary element.

For example,

n                : Spindle speed (rpm)

STL           : STereoLithography

etc.

Use "rpm" instead of "r/min".

Comment 6:

Conclusions needs to be improved.

It is necessary to more clearly show the novelty of the article and the advantages of the proposed method. Add qualitative and quantitative results of your work. What is the difference from previous work in this area? Show practical relevance.

Round 2

Reviewer 2 Report

The authors have done a good job of improving the article. The article may be accepted for publication. My congratulations to the authors. And I thank the editor for the opportunity to review the article.